# Enhancing Output Power of Textured Silicon Solar Cells by Embedding Indium Plasmonic Nanoparticles in Layers within Antireflective Coating

**DOI:** 10.3390/nano8121003

**Published:** 2018-12-04

**Authors:** Wen-Jeng Ho, Jheng-Jie Liu, Yun-Chieh Yang, Chun-Hung Ho

**Affiliations:** 1Department of Electro-Optical Engineering, National Taipei University of Technology, Taipei 10608, Taiwan; jjliu@mail.ntut.edu.tw (J.-J.L.); t105658043@ntut.edu.tw (Y.-C.Y.); 2Realtek Semiconductor Corp., No. 2, Innovation Road II, Hsinchu Science Park, Hsinchu 300, Taiwan; cunhonho@gmail.com

**Keywords:** indium nanoparticles (In NPs), textured silicon solar cells, antireflective coating (ARC), plasmonic forward scattering

## Abstract

In this study, we sought to enhance the output power and conversion efficiency of textured silicon solar cells by layering two-dimensional indium nanoparticles (In NPs) within a double-layer (SiN*_x_*/SiO_2_) antireflective coating (ARC) to induce plasmonic forward scattering. The plasmonic effects were characterized using Raman scattering, absorbance spectra, optical reflectance, and external quantum efficiency. We compared the optical and electrical performance of cells with and without single layers and double layers of In NPs. The conversion efficiency of the cell with a double layer of In NPs (16.97%) was higher than that of the cell with a single layer of In NPs (16.61%) and greatly exceeded that of the cell without In NPs (16.16%). We also conducted a comprehensive study on the light-trapping performance of the textured silicon solar cells with and without layers of In NPs within the double layer of ARC at angles from 0° to 75°. The total electrical output power of cells under air mass (AM) 1.5 G illumination was calculated. The application of a double layer of In NPs enabled an impressive 53.42% improvement in electrical output power (compared to the cell without NPs) thanks to the effects of plasmonic forward scattering.

## 1. Introduction

Most commercial solar cells are fabricated using a silicon-based wafer ranging in thickness from 150 to 200 micrometers. Light trapping in crystalline silicon solar cells is generally achieved using a pyramidal structure with an antireflective coating on the surface. This combination allows for the multiple reflection and scattering of incident light within the solar cell [1,2,3,4,5,6,7,8]. Metallic nanoparticles (NPs) [9,10,11,12] of silver (Ag NPs) [13,14,15,16], gold (Au NPs) [17,18,19,20], and aluminum (Al NPs) [21,22,23,24] have been applied to the front and/or rear-side surfaces of silicon solar cells to increase light trapping and enhance photovoltaic performance. Metallic NPs can be resonantly coupled with incident light, thereby allowing a portion of the light to be scattered into the absorber layer. Far-field forward light scattering from metallic NPs and a near-field enhanced localized field in the vicinity of the metallic NPs have been shown to boost the conversion efficiency of photovoltaic devices [9,25]. Researchers have investigated the use of various metallic materials in these devices. They have also controlled the dimensions, shapes, spacing, and surrounding dielectric environments of the NPs to enhance resonant plasmonic scattering [9,11]. In addition, a dielectric-based (TiO_2_) photonic structure using colloidal-lithographed processing was also proposed for light trapping in thin film photovoltaics [26]. Indium nanoparticles (In NPs) exhibit plasmonic resonance in the ultraviolet range (below 280 nm) and broadband plasmonic light scattering from visible and near-infrared wavelengths [27,28,29], which makes them capable of boosting the conversion efficiency of photovoltaic devices. However, there has been relatively little research on the embedding of In NP sheets within a double-layer antireflective coating (DL-ARC) to enhance the photovoltaic performance of textured silicon solar cells [30,31,32,33].

In this study, we examined the plasmonic light scattering of In NPs of various dimensions, which were embedded in a coating of SiO_2_ with a DL-ARC structure (SiN*_x_*/SiO_2_) for use in textured silicon solar cells. The plasmonic effects of samples with single and double layers of In NPs were characterized according to Raman scattering, absorbance, optical reflectance, and external quantum efficiency (EQE). We also measured the EQE and photovoltaic current-voltage (I-V) as a function of incident angle using cells with and without In NPs. We then calculated the total output electrical power of cells under AM 1.5 G illumination. The application of a double layer of In NPs enabled an impressive 53.42% improvement in output electrical power (compared to the cell without NPs) thanks to the effects of plasmonic forward scattering. The novelty of this study includes (a) the enhancement in output power and conversion efficiency of textured silicon solar cells by layering two-dimensional In NPs within a DL-ARC, and (b) a comprehensive study on the light-trapping performance of the textured silicon solar cells with and without layers of In NPs within the DL-ARC at angles from 0° to 75°, both of which are issues beyond the scope of previous studies [30,31,32,33].

## 2. Experiments

### 2.1. Characterization of Plasmonic Effects from Indium Nanoparticle Layers Embedded in SiO_2_ Coating

Quartz substrates were used as a test template to characterize the plasmonic effects of indium nanoparticles in the UV-VIS band, due to their high transparency (low absorption) at UV-band wavelengths. Figure 1 presents a schematic diagram of (a) an SiO_2_ coating (90 nm) deposited on a quartz substrate, (b) a single layer of In NPs embedded in a SiO_2_ coating (90 nm), and (c) a double layer of In NPs embedded in a SiO_2_ coating (90 nm). The SiO_2_ layer and In NPs were deposited using electron-beam (e-beam) evaporation. The In NPs were formed by depositing indium films at thicknesses of 3.8, 5, and 7 nm directly on the quartz substrate or the SiO_2_ coating and then applying rapid thermal annealing (RTA) at 200 °C under H_2_. The average surface coverage and average diameter of the In NPs were as follows: 3.8 nm (36.75% and 20.13 nm), 5 nm (41.83% and 25.03 nm), and 7 nm (46.46% and 32.14 nm). These results were calculated using image-J software from corresponding SEM images. Sample (b) was fabricated by applying a layer of In NPs on the quartz substrate and then capping it with a 90-nm coating of SiO_2_. Sample (c) was fabricated by applying an initial layer of In NPs on the quartz substrate and covering it with a 20-nm spacer layer of SiO_2_ before applying a second layer of In NPs over the spacer layer and capping it with an additional 70-nm coating of SiO_2_.

The plasmonic effects of the In NPs (single and double layers) were examined using Raman scattering and absorbance measurements. Raman scattering spectra were collected using a Raman spectrometer (UniRAM, UniNanoTech Co., Yongin-si, Giheung-gu, Korea) with a 532-nm laser (approximately 3 mW), with the signal accumulated over a period of 30 s. The observed shifts in the Raman peaks and variations in Raman signal intensity revealed that the plasmonic effects depend on the number of NP layers and the dimensions of the In NPs they comprise. Absorbance spectra were collected using a miniature spectrometer (USB4000-VIS-NIR, Ocean Optics, Inc., Largo, FL, USA) with a deuterium tungsten light source (200–2000 nm) and reflective integrating sphere (diameter of 5 cm). The obtained absorbance spectrum revealed the intensity of surface plasmon resonance (SPR) and the SPR absorption band induced by In NPs. Both measurements were also obtained from the sample without In NPs (as a control) to confirm the plasmonic effects of In NPs layered within the SiO_2_ coating.

### 2.2. Fabrication and Characterization of Plasmonic Textured Silicon Solar Cells

Boron-doped (P-type) crystalline silicon wafers with a (100) orientation and resistivity of 10 Ω-cm were cut to a thickness of 150 μm to use them as a base material for textured silicon solar cells. Following standard Radio Corporation of America (RCA) cleaning, the saw-damaged surface of the silicon wafers was removed by dipping them in a solution of H_2_O/KOH (Potassium). The surface of the wafer was then etched by dipping it in a solution of H_2_O/KOH/IPA (Isopropanol) at 80 °C for 20 min to create a surface texture in the form of randomly-arranged pyramidal structures. The textured wafers then underwent RCA cleaning prior to the application of an n^+^-Si emitter layer with a sheet resistance of approximately 80 Ω/sq using a POCl_3_ diffusion process in a tube diffusion chamber at 850 °C over a period of 3 min. The wafer was then cut into samples of 10 mm^2^. The oxide layer that formed on the surface of the samples was removed using hydrogen fluoride (HF) solution prior to the deposition of an Al film with a depth of 2 μm on the rear surface using e-beam evaporation. The as-deposited wafer was then annealed at 450 °C for 5 min to form a back electrode with a good ohmic contact to the p-silicon. Plasma-enhanced chemical vapor deposition (PECVD) was used to deposit a 70-nm silicon nitride film on the front surface as an antireflective coating. Finally, top contact grid-electrodes were formed from a Ti film (20 nm) and Al film (10,000 nm) using photolithographic etching, e-beam evaporation, and lift-off processes. The resulting textured silicon solar cells (as shown in Figure 2a) underwent characterization in terms of optical and electrical performance for use as a reference by which to evaluate the performance of the plasmonic solar cells.

Figure 2 presents schematic diagrams showing the silicon solar cells tested in this study: 2(a) presents the bare reference silicon solar cell; 2(b) shows the solar cell with an SiO_2_ coating (90 nm) without In NPs (DL-ARC; SiN*_x_*/SiO_2_); 2(c) shows the solar cell with an SiO_2_ coating (90 nm) embedded with a single layer of In NPs; 2(d) shows the solar cell with an SiO_2_ coating (90 nm) embedded with a double layer of In NPs (same dimensions in each layer) separated by a 20-nm SiO_2_ spacer layer. Note that the total thickness of the SiO_2_ coatings and embedded layers was maintained at 90 nm.

As described in Section 2.1, the layers of In NPs were fabricated by depositing indium film using e-beam evaporation with thicknesses of 3.8 nm, 5 nm, and 7 nm, followed by annealing in an RTA chamber at 200 °C under H_2_ for 30 min. A scanning electron microscope (SEM; Hitachi S-4700, Hitachi High-Tech Fielding Corporation, Tokyo, Japan) was used to characterize the sample surfaces and cross-sections. Optical reflectance (Lambda 35, PerkinElmer, Inc., Waltham, MA, USA) and external quantum efficiency (EQE; Enli Technology Co., Ltd., Kaohsiung, Taiwan) measurements were used to assess the plasmonic effects of the In NPs layers embedded within the SiO_2_ coating. The photovoltaic performance of the textured silicon solar cells (with and without In NPs layers) was assessed in terms of photovoltaic current-voltage (I-V) under AM 1.5 G illumination. The solar simulator (XES-151S, San-Ei Electric Co., Ltd., Osaka, Japan) was calibrated using a National Renewable Energy Laboratory (NREL)-certified crystalline silicon reference (PVM-894, PV Measurements Inc., Boulder, CO, USA) prior to measurement.

### 2.3. EQE and Photovoltaic Performance of Plasmonic Textured Silicon Solar Cells under Incident Light of Various Angles

We evaluated the textured silicon solar cells with and without In NP layers in terms of EQE response and photovoltaic I-V curves under illumination by an incident light source of various angles (θ), ranging from 0° to 75°, as shown in Figure 3. The light source was fixed above a stage that could be rotated from 0° to 90°. The light source was calibrated using an NREL-certified crystalline silicon reference cell at 0° prior to measurement. The output power of the cells was calculated at each incident angle to compare the total output power under daylight illumination (from AM 0700 to PM 1700). The incident angles were meant to simulate illumination at various times, as follows: 0° (noon), 45° (AM 0900/PM 1500), and 75° (AM 0700/PM 1700).

## 3. Results and Discussion

Figure 4a presents the Raman spectra of samples with the following configurations: (1) quartz substrate/SiO_2_ layer; (2) quartz substrate/SiO_2_ coating embedded with single layers of In NPs of various sizes (3.8 nm, 5 nm, and 7 nm); and (3) quartz substrate/SiO_2_ coating embedded with double layers of In NPs of various sizes (3.8 nm, 5 nm, and 7 nm). Compared to the sample with a quartz substrate/SiO_2_ coating, the samples with In NPs presented shifts in the Raman peaks at 1181, 1362, and 1485 cm^−1^. Generally, peaks in the Raman signal from metallic nanoparticles are an indication of SPR under a light source of specific wavelengths. We also observed an increase in the intensity of the Raman signals with an increase in the particle size and the number of NP layers. Thus, the most pronounced plasmonic effects were observed in the samples with a double layer of In NPs of 7 nm. Figure 4b presents the absorption spectra of samples with the following configurations: (1) quartz substrate/SiO_2_ coating; (2) quartz substrate/SiO_2_ coating embedded with single layers of In NPs of various sizes (3.8 nm, 5 nm, and 7 nm); and (3) quartz substrate/SiO_2_ coating embedded with double layers of In NPs of various sizes (3.8 nm, 5 nm, and 7 nm). The fact that the peak absorption occurred at approximately 200 nm indicates that the principal absorption of incident light occurred in the quartz substrate due to the bandgap of the quartz substrate (approximately 6 eV). Compared to the quartz substrate/SiO_2_ coating sample, the sample with a single layer of In NPs presented a higher absorption band between 220 and 300 nm, with peak absorption at approximately 260 nm. The samples with a double layer of In NPs presented far higher absorption values due to the higher density of the indium nanoparticles and the effects of light coupling between the two nanoparticle layers.

Figure 5a,b present top-view SEM images of textured silicon solar cells without and with In NPs, respectively. These images show that the minimum and maximum spacing between pyramids on the textured surface was 4 μm and 8 μm, respectively. The minimum and maximum heights were 4 and 7 μm, respectively. Figure 5c presents a particle profile of In NPs (7 nm) within the textured surface. This profile was generated from the inset of Figure 5b. The size distribution and coverage were calculated using Image-J software (National Institutes of Health, Bethesda, MD, USA). Figure 5d presents a side-view SEM image of a sample with a double layer of In NPs (7 nm) embedded within the SiO_2_ coating on a GaAs substrate. The GaAs substrate was used to examine the layer(s) of indium nanoparticles embedded in the SiO_2_ coating due to the ease with which it can be cleaved to a strip-bar for side-view SEM examination. In this 2D profile, it is easy to differentiate the first and second layers of indium nanoparticles within the SiO_2_ coating.

Figure 6a presents the optical reflectance of the reference textured silicon solar cell (Ref. Cell), the cell with an SiO_2_ coating without In NPs (ARC Cell), and cells with an SiO_2_ coating embedded with a single layer of indium nanoparticles of 3.8, 5 and 7 nm (SL-NPs Cell). The average weighted reference (*R*_W_) was calculated from the wavelength range of 380–1000 nm, as listed in Table 1. For the sake of clarity, we calculated the *R*_W_ of the cells as follows:(1)$$R_{W}=\frac{\int_{380\;\mathrm{nm}}^{1000\;\mathrm{nm}}R(\lambda )\varphi_{\mathrm{ph}}(\lambda )d\lambda }{\int_{380\;\mathrm{nm}}^{1000\;\mathrm{nm}}\varphi_{\mathrm{ph}}(\lambda )d\lambda }\times 100\%$$
where *R*(*λ*) is the optical reflectance at a given wavelength (*λ*) and *φ*_ph_(*λ*) is the photon flux of AM 1.5 G at that wavelength (*λ*). The *R*_W_ of the cells with In NPs was lower than that of the reference cell due to SPR absorption in the wavelength range of 200–350 nm (Figure 4b) and plasmonic forward scattering beyond 600 nm, both of which were induced by the In NPs. The low *R*_W_ indicates that the NPs enabled more of the incident light to be trapped in the silicon. Samples with larger nanoparticles (7 nm) presented lower *R*_W_ values (3.34%) than the samples with 3.8-nm nanoparticles (3.78%). Again, we can see that larger In NPs were able to trap more of the incident light. We therefore fabricated samples with two layers of larger In NPs (7 nm) for further study and comparison. Figure 6b presents the optical reflectance of the reference cell, the cell with an SiO_2_ coating (no NPs), and cells with single and double layers of In NPs of 7 nm (DL-NPs Cell). We calculated the *R*_W_ of all tested cells over a wavelength range of 380–1000 nm, the results of which are listed in Table 1. The lowest *R*_W_ value (2.32%) was obtained from the cell with the double layer of In NPs embedded within the SiO_2_ coating.

Figure 7a presents the EQE response of the reference solar cell, the cell with an SiO_2_ coating (no In NPs), and cells with a single layer of In NPs of various sizes (3.8, 5, and 7 nm) embedded in an SiO_2_ coating. The EQE values of cells with In NPs were higher than those without In NPs across the entire wavelength range, due to the effects of plasmonic forward scattering induced by the NPs. The EQE values of cells with larger NPs were slightly higher than those with smaller NPs. The EQE response values are in good agreement with the optical reflectance results. Figure 7b presents the EQE response of the reference cell, the cell with a SiO_2_ coating (no In NPs), and cells with either a single layer of In NPs (7 nm) or a double layer of In NPs (7 nm). For the sake of clarity, we calculated the average weighted EQE (*EQE*_W_) of the cells as follows:(2)$$EQE_{W}=\frac{\int_{380\;\mathrm{nm}}^{1000\;\mathrm{nm}}EQE(\lambda )\varphi_{\mathrm{ph}}(\lambda )d\lambda }{\int_{380\;\mathrm{nm}}^{1000\;\mathrm{nm}}\varphi_{\mathrm{ph}}(\lambda )d\lambda }\times 100\%$$
where *EQE(λ)* is the *EQE* at a given wavelength *(λ)* and *φ*_ph_
*(λ)* is the photon flux of AM 1.5 G at that wavelength *(λ)*. The *EQE*_W_ values were as follows: double layer of 7-nm In NPs (92.74%), single layer of 7-nm In NPs (91.35%), SiO_2_ coating without NPs (88.97%), and reference cell (88.82%). Table 1 summarizes the *EQE*_W_ of the cells calculated over a wavelength range of 380–1000 nm. The EQE values of cells with double layers of In NPs exceeded those of cells with a single layer due to the higher density of NPs and more pronounced plasmonic forward scattering.

Figure 8a presents the photovoltaic J-V curves obtained from the reference cell, the cell with an SiO_2_ coating (no In NPs), and cells with a single layer of In NPs of various sizes (3.8, 5, and 7 nm) under normal incident illumination (θ = 0°). The short-circuit current densities (*J*_sc_) and conversion efficiency (η) values were as follows: single layer of 7-nm NPs (40.26 mA/cm^2^ and 16.61%), single layer of 5-nm NPs (39.95 mA/cm^2^ and 16.51%), single layer of 3.8-nm NPs (39.77 mA/cm^2^ and 16.44%), SiO_2_ coating without NPs (39.61 mA/cm^2^ and 16.34%), and reference cell (39.19 mA/cm^2^ and 16.16%). The *J*_sc_ values of cells with NPs were higher due to plasmonic forward scattering than those without NPs. The *J*_sc_ values of cells with larger NPs were slightly higher than those of cells with smaller NPs, due to stronger plasmonic forward scattering.

Figure 8b presents the photovoltaic J-V curves of cells with a single layer of 7-nm In NPs and a double layer of 7-nm In NPs. The photovoltaic performance of the proposed cells is summarized in Table 2. Adding the second layer of In NPs increased the *J*_sc_ value from 40.26 to 40.92 mA/cm^2^, and the η value from 16.61% to 16.97%, compared to a single layer of In NPs (7 nm). Adding two layers of In NPs (7 nm) increased the *J*_sc_ value from 39.61 to 40.92 mA/cm^2^, and the η value from 16.34% to 16.97%, compared to DL-ARC without In NPs. These results demonstrate that using larger In NPs and including multiple layers of In NPs facilitates the trapping of incident light and enhances *J*_sc_ and η, due to stronger plasmonic forward scattering.

Figure 9 presents the *EQE*_W_ and *J*_sc_ of the reference cell, the cell with an SiO_2_ coating (no NPs), and cells with single and double layers of In NPs under incident angles of 0°–75°. Increasing the incident angle resulted in a gradual decrease in *EQE*_W_ and *J*_sc_ values in all tested cells. Compared to cells without NPs, we obtained higher *EQE*_W_ and *J*_sc_ values from cells with single and double layers of In NPs at all incident angles. At a high incident angle of 75°, the double layer cells presented a *J*_sc_ decrement of 25.2% (from 40.92 to 30.06 mA/cm^2^), compared to the decrement of 76.3% (from 39.61 to 9.38 mA/cm^2^) from cells without NPs. Overall, *J*_sc_ was proportional to EQE and η was proportional to *J*_sc_ in all of the photovoltaic devices. This means that a higher *J*_sc_ would no doubt result in a higher electrical output, as well as a higher conversion efficiency.

Figure 10a presents the calculated electrical output power of all evaluated solar cells under illumination from −75° to 0° (sun rising; i.e., AM 0700 to noon) and then from 0° to 75° (sun descending; i.e., noon to PM 1700). At all illumination times/angles, the output power of cells with a double layer of In NPs exceeded that of cells with a single layer and cells without NPs. Figure 10b presents the daily output energy of all evaluated solar cells. For the sake of clarity, we calculated the electrical output power (*P*_E_) and the daily output energy (*E*_day_) of the cells as follows:(3)$$P_{E}=V_{\mathrm{oc}}\times J_{\mathrm{sc}}\times FF$$
where *V*_oc_ is the open-circuit voltage, *J*_sc_ is the short-circuit current density, and *FF* is the fill factor. The total *P*_E_ values of cells with an area of 10 mm^2^ were as follows: double layer of 7-nm NPs (141.38 mW), single layer of 7-nm NPs (114.13 mW), SiO_2_ coating (94.92 mW), and reference cell (92.15 mW).
(4)$$E_{day}=\sum_{i=AM0700}^{i=PM1700}{\left(P_{E}\right)}_{i}\times 1_{}\;hour$$
the *E*_day_ values of the cell with an SiO_2_ coating and the reference cell were 94.92 and 92.15 mW·h, respectively. Using these as reference values, the inclusion of a single layer of In NPs (7 nm) increased *E*_day_ by 20.24% and 23.85%, respectively. The inclusion of a double layer of In NPs (7 nm) increased *E*_day_ by 48.95% and 53.42%, respectively.

## 4. Conclusions

In this study, we examined the light trapping effects of In NPs according to optical reflectance and EQE measurements, with a particular focus on the dimensions of the NPs and the number of layers of NPs. A double layer of In NPs within the antireflective coating resulted in pronounced plasmonic forward scattering, which greatly enhanced the output power and conversion efficiency of the textured silicon solar cells. The inclusion of a double layer of In NPs increased the conversion efficiency from 16.16% to 16.97%, compared to the reference cell without In NPs. We also examined the light-trapping performance of cells with and without In NPs at incidence angles from 0° to 75°. At all angles, the output power delivered from cells with a double layer of In NPs exceeded that of cells with a single layer and those without NPs. The cumulative output power (one day) delivered by the cell with an area of 10 mm^2^ with a double layer of In NPs was 141.38 mW, which greatly exceeds the 94.92 mW of the cell without In NPs.

## Figures and Tables

**Figure 1 nanomaterials-08-01003-f001:**
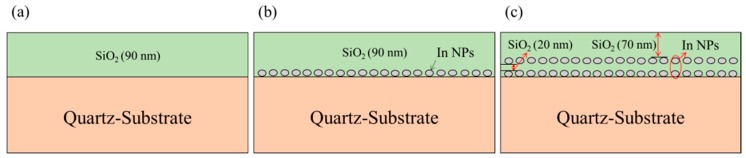
Schematic diagrams of proposed samples: (**a**) SiO_2_ coating (90 nm) deposited on quartz substrate; (**b**) single layer of In NPs embedded in SiO_2_ coating (90 nm) on quartz substrate; and (**c**) double layer of In NPs embedded in SiO_2_ coating (90 nm).

**Figure 2 nanomaterials-08-01003-f002:**
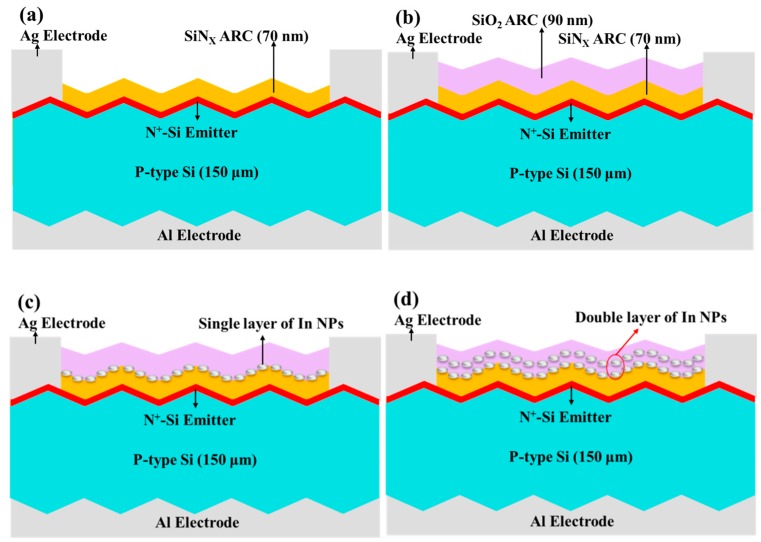
Schematic illustrations showing cells evaluated in this study: (**a**) reference textured silicon solar cell; (**b**) reference cell with SiO_2_ coating (90 nm); (**c**) reference cell with SiO_2_ coating (90 nm) embedded with single layer of In NPs; and (**d**) reference cell with SiO_2_ coating (90 nm) embedded with double layer of In NPs.

**Figure 3 nanomaterials-08-01003-f003:**
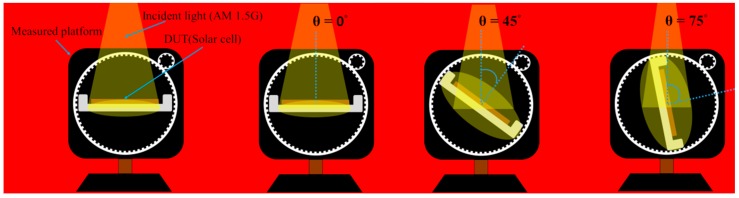
Varying incident light source angle (θ) while illuminating textured silicon solar.

**Figure 4 nanomaterials-08-01003-f004:**
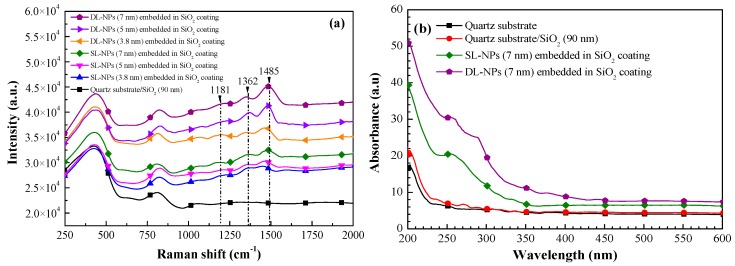
(**a**) Raman spectra; (**b**) absorption spectra of all tested samples.

**Figure 5 nanomaterials-08-01003-f005:**
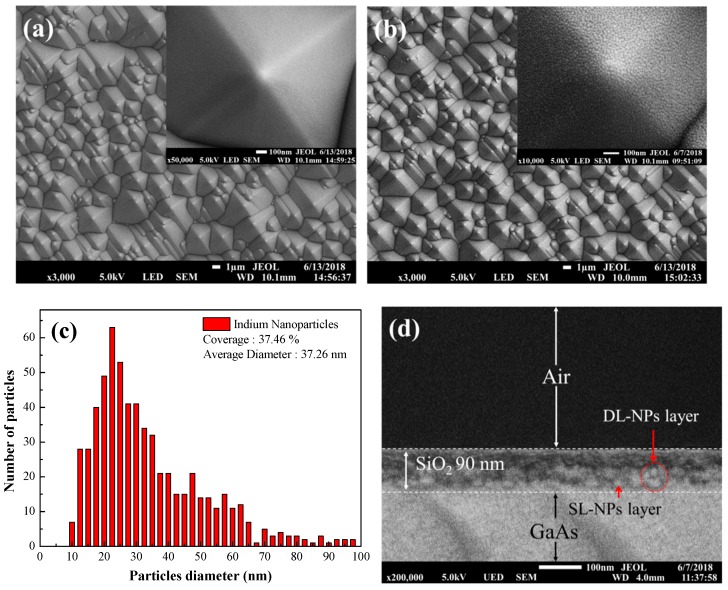
Top-view SEM images of textured silicon solar cells: (**a**) without and (**b**) with In NPs; (**c**) particle profile of In NPs (7 nm) within textured surface; (**d**) side-view SEM image of double layer of In NPs (7 nm) embedded within SiO_2_ coating on GaAs substrate. The inset in Figure 5a,b is an enlarge graph of a pyramid structure on the cells without and with In NPs, respectively.

**Figure 6 nanomaterials-08-01003-f006:**
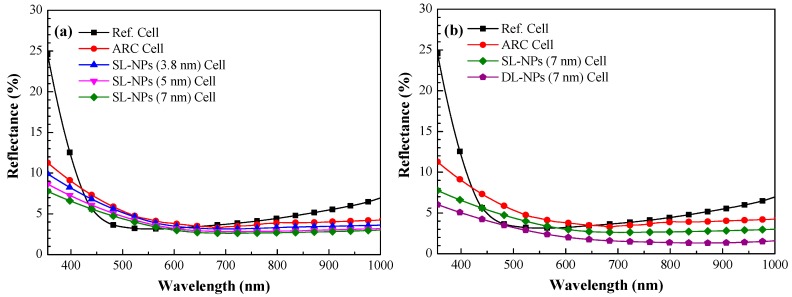
Optical reflectance: (**a**) reference cell, cell with SiO_2_ coating (no In NPs), cell with single layer of In NPs (3.8 nm, 5 nm, and 7 nm) embedded in SiO_2_ coating; (**b**) cell with double layer of In NPs (7 nm) embedded in SiO_2_ coating.

**Figure 7 nanomaterials-08-01003-f007:**
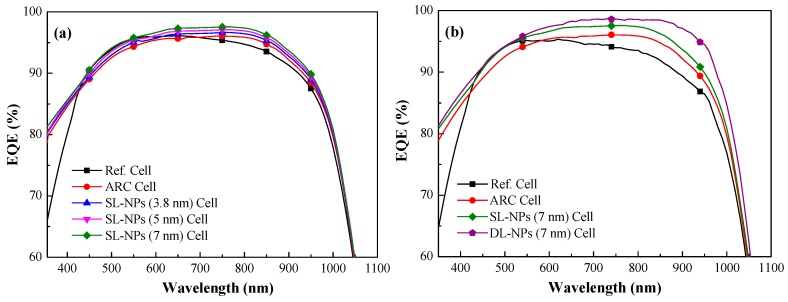
EQE response values: (**a**) reference cell, cell with SiO_2_ coating (no NPs), and cell with single layer of In NPs (3.8 nm, 5 nm, and 7 nm); (**b**) reference cell, cell with SiO_2_ coating (no NPs), cell with single layer of In NPs (7 nm), and cell with double layer of In NPs (7 nm).

**Figure 8 nanomaterials-08-01003-f008:**
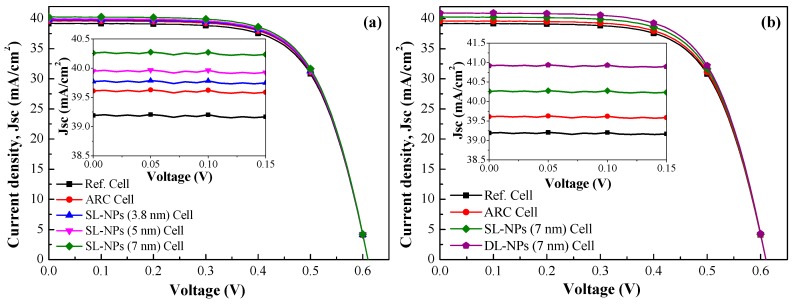
Photovoltaic I-V curves: (**a**) reference cell, cell with SiO_2_ coating (no NPs), and cell with single layer of In NPs (3.8 nm, 5 nm, and 7 nm); (**b**) reference cell, cell with SiO_2_ coating (no NPs), and cell with double layer of In NPs (7 nm). The inset in Figure 8a,b is an enlarge graph of *J*_sc_ of all evaluated cells at voltage 0−0.15 V.

**Figure 9 nanomaterials-08-01003-f009:**
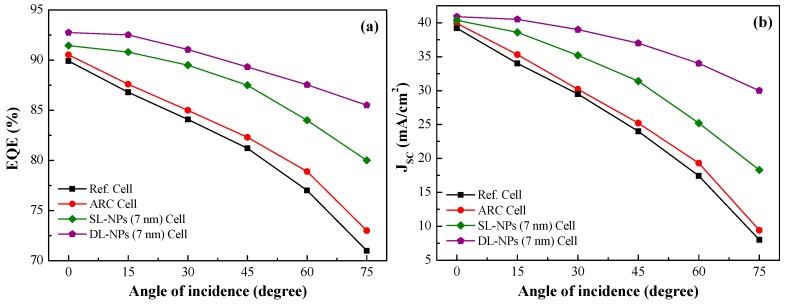
(**a**) EQE and (**b**) *J*_sc_ as a function of incident angle of reference cell, cell with SiO_2_ coating (no NPs), cell with single layer of In NPs (7 nm), and cell with double layer of In NPs (7 nm).

**Figure 10 nanomaterials-08-01003-f010:**
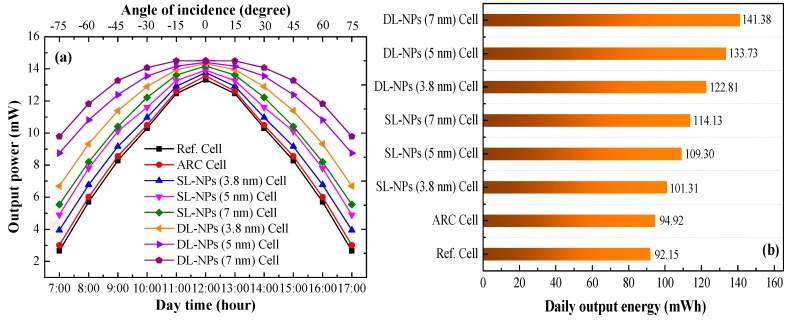
(**a**) Output electrical power; (**b**) output electrical energy of all tested solar cells under illumination at incidence angles from −75° to 0° (sun rising) and from 0° to 75° (sun descending).

**Table 1 nanomaterials-08-01003-t001:** Average Weighted Reference (*R*_W_) and Average Weighted External Quantum Efficiency (*EQE_W_*) of Proposed Cells.

Silicon Solar Cell	*R*_W_ (%) @ 380–1000 nm	*EQE*_W_ (%) @ 380–1000 nm
Ref. Cell	4.58	88.82
DL-ARC	4.15	89.87
SL-In NPs (3.8 nm) Cell	3.78	90.13
SL-In NPs (5 nm) Cell	3.54	90.45
SL-In NPs (7 nm) Cell	3.34	91.35
DL-In NPs (7 nm) Cell	2.32	92.74

**Table 2 nanomaterials-08-01003-t002:** Photovoltaic Performance of Proposed Cells Under AM 1.5 G Illumination at Normal Incidence.

Silicon Solar Cell	*J*_sc_ (mA/cm^2^)	*V*_oc_ (mV)	Fill Factor (%)	η (%)	Δ*J*_sc_ (%)	Δη (%)
Ref. Cell	39.19	609.40	67.68	16.16	---	---
ARC Cell	39.61	609.50	67.70	16.34	1.07	1.11
SL-In NPs (3.8 nm) Cell	39.77	609.51	67.85	16.44	1.47	1.73
SL-In NPs (5 nm) Cell	39.95	609.52	67.84	16.51	1.93	2.16
SL-In NPs (7 nm) Cell	40.26	609.53	67.72	16.61	2.73	2.78
DL-In NPs (7 nm) Cell	40.92	609.50	68.05	16.97	4.41	5.01

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
