# Peer review of "Enhancing Output Power of Textured Silicon Solar Cells by Embedding Indium Plasmonic Nanoparticles in Layers within Antireflective Coating"

_nanomaterials, 2018, doi:10.3390/nano8121003_

Reviewer 1 Report

The topic of he manuscript (indium related plasmonic effects on solar cells efficiency) was very well/extensively studied by the authors (even the titles are very close):

1. Performance of plasmonic silicon solar cells using indium nanoparticles deposited on a patterned TiO2 matrix, Thin Solid Films 570:194-199 · November 2014

2. Plasmonic Light Scattering in Textured Silicon Solar Cells with Indium Nanoparticles from Normal to Non-Normal Light Incidence Wen-Jeng Ho et al, Materials (Basel). 2017 Jul; 10(7): 737.

3. Performance-Enhanced Textured Silicon Solar Cells Based on Plasmonic Light Scattering Using Silver and Indium Nanoparticles by Wen-Jeng Ho et al, Materials 2015, 8, 6668–667

4. Performance Enhancement of Plasmonics Silicon Solar Cells Using Al2O3/In NPs/TiO2 Antireflective Surface Coating, by Wen-Jeng Ho et al  Applied Surface Science 354 · March 2015

5. Photovoltaic performance enhancement of plasmonics silicon solar cells using indium nanoparticles embedded in Al2O3/TiO2 layer structure by Wen-Jeng Ho et al IEEE International Nanoelectronics Conference (INEC), July 2014

I do not see them in the references list. Why?

It is not clear for me what is the novelty, the contribution to the development of current scientific interest and the interest to the readers.

The authors have to clarify in a very convincing manner this point… otherwise it seems as a small variation of the previous papers 

Author Response

Comments and Suggestions for Authors

The topic of the manuscript (indium related plasmonic effects on solar cells efficiency) was very well/extensively studied by the authors (even the titles are very close):

1. Performance of plasmonic silicon solar cells using indium nanoparticles deposited on a patterned TiO2 matrix, Thin Solid Films 570:194-199 · November 2014

2. Plasmonic Light Scattering in Textured Silicon Solar Cells with Indium Nanoparticles from Normal to Non-Normal Light Incidence Wen-Jeng Ho et al, Materials (Basel). 2017 Jul; 10(7): 737.

3. Performance-Enhanced Textured Silicon Solar Cells Based on Plasmonic Light Scattering Using Silver and Indium Nanoparticles by Wen-Jeng Ho et al, Materials 2015, 8, 6668–667

4. Performance Enhancement of Plasmonics Silicon Solar Cells Using Al2O3/In NPs/TiO2 Antireflective Surface Coating, by Wen-Jeng Ho et al  Applied Surface Science 354 · March 2015

5. Photovoltaic performance enhancement of plasmonics silicon solar cells using indium nanoparticles embedded in Al2O3/TiO2 layer structure by Wen-Jeng Ho et al IEEE International Nanoelectronics Conference (INEC), July 2014

I do not see them in the references list. Why?

Ans:

(1) The authors would like to express their sincere appreciation to the reviewer for useful comments.

(2) In this study, we sought to enhance the output power and conversion efficiency of textured silicon solar cells by layering two-dimensional indium nanoparticles (In NPs) within a double-layer (SiNx/SiO2) antireflective coating (ARC) due to plasmonic forward scattering.

(3) Thanks your remark that the above-mentioned papers (#3 and #4) about indium NPs embedded in ARC related plasmonic effects on solar cells efficiency are cited, on the revised manuscript.

(4) The new cited papers are #32 and #33 listed in the reference list, on the revised manuscript.

It is not clear for me what is the novelty, the contribution to the development of current scientific interest and the interest to the readers. The authors have to clarify in a very convincing manner this point… otherwise it seems as a small variation of the previous papers 

Ans:

(1) In this study, we sought to enhance the output power and conversion efficiency of textured silicon solar cells by layering two-dimensional indium nanoparticles (In NPs) within a double-layer (SiNx/SiO2) antireflective coating (ARC) due to plasmonic forward scattering. We compared the optical and electrical performance of cells with and without single layers and double layers of In NPs. We also conducted a comprehensive study on the light-trapping performance of the textured silicon solar cells with and without layers of In NPs within the double layer of ARC at angles from 0° to 75°.

       (2) The novelty of this study was: (a) the enhanced in output power and conversion 

          efficiency of textured silicon solar cells by layering two-dimensional indium 

          nanoparticles (In NPs) within a double-layer (SiNx/SiO2) antireflective coating  

         (ARC), (b) a comprehensive study on the light-trapping performance of the textured 

          silicon solar cells with and without layers of In NPs within the double layer of ARC 

          at angles from 0° to 75°, both are beyond the previous studies.

Reviewer 2 Report

The manuscript titled, "Enhancing output power and conversion efficiency of textured silicon solar cells by embedding indium plasmonic nanoparticles in layers within antireflective coating" auhtored by Wen-Jeng Ho et al , reports the  application of plasmonic In NPs in a anti-reflective coating to enhance the output power and conversion efficiency of Si solar cells. The proposed work is interesting and will be helpful to the researchers working in the field of Si solar cells. Authors have successfully applied plasmonic  In NPs in anti reflection coating and studied the conversion efficiency. Results and discussion are sound and well presented. The current manuscript is suitable for publication in Nanomaterials journal.

Authors must fix the following issues before publication.

01. Insert the scale bars in Figure 5 for SEM images.

02. The mechanism for the improvement in In NPs modified solar cells efficiency should be highlighted in the discussions part. 

Author Response

Comments and Suggestions for Authors

The manuscript titled, "Enhancing output power and conversion efficiency of textured silicon solar cells by embedding indium plasmonic nanoparticles in layers within antireflective coating" auhtored by Wen-Jeng Ho et al., reports the application of plasmonic In NPs in an anti-reflective coating to enhance the output power and conversion efficiency of Si solar cells. The proposed work is interesting and will be helpful to the researchers working in the field of Si solar cells. Authors have successfully applied plasmonic In NPs in antireflection coating and studied the conversion efficiency. Results and discussion are sound and well presented. The current manuscript is suitable for publication in Nanomaterials journal.

Ans:

The authors would like to express their sincere appreciation to the reviewer for useful comments. Thanks.

Authors must fix the following issues before publication.

01. Insert the scale bars in Figure 5 for SEM images.

Ans:

The scale bars in Figure 5 for SEM images have been revised for more clearly, on the revised manuscript.

02. The mechanism for the improvement in In NPs modified solar cells efficiency should be highlighted in the discussions part. 

Ans:

The mechanism of plasmonic forward scattering of In NPs was the performance improved factor in plasmonic solar cells efficiency.

We revised asThe Jsc values of cells with NPs were higher due to plasmonic forward scattering than those without NPs. The Jsc values of cells with larger NPs were slightly higher than those of cells with smaller NPs, due to stronger plasmonic forward scattering” and “These results demonstrate that using larger In NPs and including multiple layers of In NPs facilitates the trapping of incident light and enhances Jsc and η due to having stronger plasmonic forward scattering”, on the revised manuscript.

Reviewer 3 Report

The paper titled Enhancing output power and conversion efficiency of textured silicon solar cells by embedding indium plasmonic nanoparticles in layers within antireflective coating, from Wen-Jeng Ho et al deals with today topic that start being looking at 45 years ago, which it is how we can improve solar cells efficiency? One possible way (but full limited to materials conversation boundaries) is by improving the ARC, exploiting the possible enhancement of light pathway and reducing so reflecting losses. Today there are several other better strategies to do so, but even so this could be a good contribution as far as science is concerned. The paper however needs some mandatory corrections and needs to be more open as far as the state of the at is concerned, aiming to capture the attention of a broader audience. Please see below my comments.

Title: The title is too long. Please just try to capture the key f the study that concerns the introduction of conductive indium nanoparticles on the ARC (what?) to enhance the solar cell efficiency of textured silicon solar cells.

Abstract: It contains the key indicators of the study performed, where it is missing information about the structure in which the ARC is based

Introduction: Worthwhile also to mention the use photonic nanostructures for solar cell light trapping, as the ones based on TiO2 (see O. Sanches-Sobrado et al in JOURNAL OF MATERIALS CHEMISTRY C, Vol. 5 pp. 6852-6861, JUL 21 2017), as a clear alternative to Indium.

Experiments: Are well described missing information about the number of samples/devices tested; what are the statistics of the data evaluated, what it is the standard deviation observed and the errors associated to to data depicted. Please, also specify the environment conditions where the solar cells were tested and if you notice any degradation of data collected with time.

Results and Discussion: The discussion is well performed and consistent with the experimental achievements.

Conclusions: Are clear and focus the key set of achievements obtained

References: need some upgrade

Author Response

Comments and Suggestions for Authors

The paper titled Enhancing output power and conversion efficiency of textured silicon solar cells by embedding indium plasmonic nanoparticles in layers within antireflective coating, from Wen-Jeng Ho et al deals with today topic that start being looking at 45 years ago, which it is how we can improve solar cells efficiency? One possible way (but full limited to materials conversation boundaries) is by improving the ARC, exploiting the possible enhancement of light pathway and reducing so reflecting losses. Today there are several other better strategies to do so, but even so this could be a good contribution as far as science is concerned. The paper however needs some mandatory corrections and needs to be more open as far as the state of the at is concerned, aiming to capture the attention of a broader audience. Please see below my comments.

Ans:

The authors would like to express their sincere appreciation to the reviewer for useful comments.

Title: The title is too long. Please just try to capture the key f the study that concerns the introduction of conductive indium nanoparticles on the ARC (what?) to enhance the solar cell efficiency of textured silicon solar cells.

Ans:

We revised the title as “Enhancing output power of textured silicon solar cells by embedding indium plasmonic nanoparticles in layers within antireflective coating”, on the revised manuscript.

Abstract: It contains the key indicators of the study performed, where it is missing information about the structure in which the ARC is based.

Ans:

The structure of ARC, in this work, is consisted of a SiNx layer and SiO2 layer. Thus, we revised asIn this study, we sought to enhance the output power and conversion efficiency of textured silicon solar cells by layering two-dimensional indium nanoparticles (In NPs) within a double-layer (SiNx/SiO2) antireflective coating (ARC) to induce plasmonic forward scattering”, on the revised manuscript.

Introduction: Worthwhile also to mention the use photonic nanostructures for solar cell light trapping, as the ones based on TiO2 (see O. Sanches-Sobrado et al in JOURNAL OF MATERIALS CHEMISTRY C, Vol. 5 pp. 6852-6861, JUL 21 2017), as a clear alternative to Indium.

Ans:

We cited this paper as reference #26 and revised as “In addition, dielectric-based (TiO2) photonic structure using colloidal-lithographed processing was also proposed for light trapping in thin film photovoltaics [26] , on the revised paper

Experiments: Are well described missing information about the number of samples/devices tested; what are the statistics of the data evaluated, what it is the standard deviation observed and the errors associated to to data depicted. Please, also specify the environment conditions where the solar cells were tested and if you notice any degradation of data collected with time.

Ans:

(1) All data were obtained from an average value by measuring 3 times. The statistics of the data evaluated are not present in this study.

(2) The solar cells were tested at room temperature approximately 23-25 ºC.

(3) No degradation in photovoltaic performance was observed because the devices have been passivated with a dielectric layer. 

Results and Discussion: The discussion is well performed and consistent with the experimental achievements.

Ans:

Thanks.

Conclusions: Are clear and focus the key set of achievements obtained.

Ans:

Thanks.

References: need some upgrade

Ans:

References (#26, #32, and #33) have been revised and upgraded, on the revised manuscript.

 Round  2

Reviewer 1 Report

The authors gave proper answers to the comments.

It would be very good and useful to include them into the manuscript body text.

Author Response

Comments and Suggestions for Authors

The authors gave proper answers to the comments.

Ans:

Thanks a lot.

It would be very good and useful to include them into the manuscript body text.

Ans:

The novelty of this study was added into Introduction Section.

We revised as “In this study, we examined the plasmonic light scattering of In NPs of various dimensions, which were embedded in a coating of SiO2 with a DL-ARC structure (SiNx/SiO2) for use in textured silicon solar cells. The plasmonic effects of samples with single and double layers of In NPs were characterized according to Raman scattering, absorbance, optical reflectance, and external quantum efficiency (EQE). We also measured the EQE and photovoltaic current-voltage (I-V) as a function of incident angle using cells with and without In NPs. We then calculated the total output electrical power of cells under AM 1.5 G illumination. The application of a double layer of In NPs enabled an impressive 53.42% improvement in output electrical power (compared to the cell without NPs) thanks to the effects of plasmonic forward scattering. The novelty of this study includes (a) the enhanced in output power and conversion efficiency of textured silicon solar cells by layering two-dimensional In NPs within a DL-ARC, (b) a comprehensive study on the light-trapping performance of the textured silicon solar cells with and without layers of In NPs within the DL-ARC at angles from 0° to 75°, both issues are beyond the previous studies [30-33]”, on the revised manuscript.
